# Genetic Association of the Functional *WDR4* Gene in Male Fertility

**DOI:** 10.3390/jpm11080760

**Published:** 2021-07-30

**Authors:** Yu-Jia Wang, Eko Mugiyanto, Yun-Ting Peng, Wan-Chen Huang, Wan-Hsuan Chou, Chi-Chiu Lee, Yu-Shiuan Wang, Lalu Muhammad Irham, Dyah Aryani Perwitasari, Ming-I Hsu, Wei-Chiao Chang

**Affiliations:** 1Department of Clinical Pharmacy, School of Pharmacy, College of Pharmacy, Taipei Medical University, Taipei 11031, Taiwan; jia_925@hotmail.com (Y.-J.W.); 101353@w.tmu.edu.tw (Y.-T.P.); ocean.chou@tmu.edu.tw (W.-H.C.); lalu.irham@pharm.uad.ac.id (L.M.I.); 2PhD Program in Clinical Drug Development of Chinese Herbal Medicine, College of Pharmacy, Taipei Medical University, Taipei 11031, Taiwan; giyan77@gmail.com (E.M.); yswang1004@gmail.com (Y.-S.W.); 3Department of Pharmacy, Wan Fang Hospital, Taipei Medical University, Taipei 11696, Taiwan; 4Institute of Cellular and Organismic Biology, Academia Sinica, Taipei 115, Taiwan; wanchen.huang@gmail.com (W.-C.H.); ccjosephlee@gmail.com (C.-C.L.); 5Master Program for Clinical Pharmacogenomics and Pharmacoproteomics, School of Pharmacy, Taipei Medical University, Taipei 11031, Taiwan; 6Faculty of Pharmacy, University of Ahmad Dahlan, Yogyakarta 55164, Indonesia; diahperwitasari2003@yahoo.com; 7Department of Obstetrics and Gynaecology, Wan Fang Hospital, Taipei Medical University, Taipei 11696, Taiwan; 8Integrative Research Center for Critical Care, Wan Fang Hospital, Taipei Medical University, Taipei 11696, Taiwan

**Keywords:** *WDR4*, genetic variants, male fertility, sperm quality

## Abstract

Infertility is one of the important problems in the modern world. Male infertility is characterized by several clinical manifestations, including low sperm production (oligozoospermia), reduced sperm motility (asthenozoospermia), and abnormal sperm morphology (teratozoospermia). *WDR4,* known as Wuho, controls fertility in Drosophila. However, it is unclear whether *WDR4* is associated with clinical manifestations of male fertility in human. Here, we attempted to determine the physiological functions of *WDR4* gene. Two cohorts were applied to address this question. The first cohort was the general population from Taiwan Biobank. Genomic profiles from 68,948 individuals and 87 common physiological traits were applied for phenome-wide association studies (PheWAS). The second cohort comprised patients with male infertility from Wan Fang Hospital, Taipei Medical University. In total, 81 male participants were recruited for the genetic association study. Clinical records including gender, age, total testosterone, follicle-stimulating hormone (FSH), luteinizing hormone (LH), total sperm number, sperm motility, and sperm morphology were collected. In the first cohort, results from PheWAS exhibited no associations between *WDR4* genetic variants and 87 common physiological traits. In the second cohort, a total of four tagging single-nucleotide polymorphisms (tSNPs) from *WDR4* gene (rs2298666, rs465663, rs2248490, and rs3746939) were selected for genotyping. We found that SNP rs465663 solely associated with asthenozoospermia. Functional annotations through the GTEx portal revealed the correlation between TT or TC genotype and low expression of *WDR4*. Furthermore, we used mouse embryonic fibroblasts cells from mwdr4 heterozygous (+/‒) mice for functional validation by western blotting. Indeed, low expression of *WDR4* contributed to ROS-induced DNA fragmentation. In conclusion, our results suggest a critical role of *WDR4* gene variant as well as protein expression in asthenozoospermia.

## 1. Introduction

Infertility is defined as a disease characterized by the inability of couples to conceive a pregnancy after 12 months of regular unprotected intercourse [1]. Infertility is a global health issue affecting 8~12% of couples worldwide, and male infertility factors are responsible for about 20~30% of infertility cases. Approximately 7% of the male population is affected by infertility [2]. Evaluating male infertility is mainly based on semen analyses. The most significant abnormalities of semen quality include a low sperm number (oligozoospermia), poor sperm motility (asthenozoospermia), and abnormal sperm morphology (teratozoospermia) [3,4]. Oxidative stress has been considered as a critical factor to infertility that possibly associated with the DNA fragmentation [5]. Moreover, the DNA fragmentation levels were reported to correlate with sperm motility [6].

Genetic variations are considered as one of the important factors in male infertility. For example, the azoospermia factor (AZF) region in the long arm of the Y chromosome contains three sub-regions referred to as AZFa, AZFb, and AZFc. Microdeletions in these sub-regions cause impaired sperm production [7]. In addition, HSFY, which is related to the heat-shock transcription factor (HSF) family, has a long open reading frame carrying an HSF-type DNA binding domain involved in azoospermia. Some point mutations in the testis-specific HSFY gene family, such as microdeletions, may be a cause of unexplained cases of idiopathic male infertility [1]. Furthermore, Ozdemir et al. evaluated the genetic variants of SRY and AZF in male infertility [8]. Moreover, three SNPs (rs4920566, rs11763979, and rs3741843) on taste receptor genes (TASR) are associated with male infertility [9]. Several genes were further identified to correlate with infertility-related azoospermia in humans, such as *RNU7-6P*, *ZFP64* [10], *SOX5*, *PLCH2* [11], and *ART3* genes [12].

Wu et al. reported that the *wh* gene (*Wuho* gene with WD40 repeats, also named *WDR4* in homo sapiens) is important in *Drosophila* spermatogenesis [13]. However, the physiological functions as well as the clinical manifestation for genetic variations of *WDR4* gene are still unclear. Here, two cohorts were applied to address this question. The first cohort was the general population from Taiwan Biobank used for phenome-wide association study (PheWAS). The association between *WDR4* variants and 87 common traits was comprehensively screened. The second cohort was male participants recruited from Wan Fang Hospital. The effects of *WDR4* variants on human male infertility (oligozoospermia, asthenozoospermia, teratozoospermia) were investigated.

## 2. Materials and Methods

### 2.1. Participant Recruitment and Sample Collection

Men aged ≥ 20 years old were recruited from the infertility clinic of Wan Fang Hospital, Taipei Medical University between November 2013 and June 2015. Participants with a history of vasectomy or cancer were excluded from the study. Semen samples were collected from every participant by masturbation after 3~5 days of sexual abstinence. These samples were kept at room temperature and delivered to the reproductive laboratory for semen analysis within 1 h. Peripheral blood samples were collected for serum hormone analysis and genomic DNA extraction. The study protocol was approved by the Taipei Medical University-Joint Institutional Review Board (TMU-JIRB) (No: 201302040), and written informed consent was received from all participants.

### 2.2. Semen Analysis

Semen samples were examined after liquefaction. Semen analysis was conducted separately by two experienced technicians in the reproductive laboratory. Semen parameters including total sperm number, sperm motility, and sperm morphology were evaluated. The World Health Organization (WHO) 2010 recommendations for semen analysis were followed [14]. A semen sample (10 µL) was first transferred to a MAKLER counting chamber (Irvine Scientific, Santa Ana, CA, USA) to measure total number of spermatozoa. Oligozoospermia was defined as a number of sperm < 39 × 10^6^ spermatozoa. The motility of each spermatozoon was graded into three categories: progressive motility (PR), non-progressive motility (NP), and immotile spermatozoa. Asthenozoospermia was defined as PR < 32%. For sperm morphological evaluation, each spermatozoon was classified into eight groups as follows: normal sperm (NS), amorphous head (AH), duplicated head (DH), large head (LH), small head (SH), normal head with other defect (NH), tapering head (TH), and other defect (OD). Teratozoospermia was defined as <4% of morphologically normal spermatozoa. In addition, combined oligozoospermia was defined as participants with sperm number less than 39 × 10^6^ spermatozoa and other defects (astheno- and/or teratozoospermia), a rationale also applied to two other sperm characteristics (combined asthenozoospermia and combined teratozoospermia).

### 2.3. Genomic DNA Extraction

Peripheral blood samples were first centrifuged at 3000 rpm and 4 °C for 10 min to separate serum and blood cells. The buffy coat layer was extracted and washed with red blood cell (RBC) lysis buffer to isolate peripheral blood mononuclear cells (PBMCs). PBMCs were then lysed using a cell lysis buffer. Proteins were precipitated using a protein precipitation solution followed by 95% isopropanol and 80% alcohol to isolate total genomic DNA. Finally, the DNA purity and concentration were measured using NanoDrop (Thermo Fisher Scientific, Wilmington, DE, USA).

### 2.4. Genotyping of SNPs in the WDR4 Gene

Tagging (t)SNPs with a minimum allele frequency (MAF) of >10% in a Beijing Han Chinese (CHB) population were selected through UCSC (http://genome.ucsc.edu) and HapMap vers. 2010-08_phase II + III (http://hapmap.ncbi.nlm.hig.gov/). Haploview 4.2 was applied for tSNP selection. A total of four SNPs of the *WDR4* gene were selected and genotyped (Figure 1). Characteristics of these SNPs are shown in Appendix A. Genotyping was performed using the TaqMan Allelic Discrimination Assay (Applied Biosystems, Foster City, CA, USA). A polymerase chain reaction (PCR) was carried out with an ABI StepOnePlus Thermal Cycler (Applied Biosystems). The fluorescence from different probes was measured and analyzed by System SDS software vers. 2.2.2 (Applied Biosystems).

### 2.5. Functional Annotation Data Query

Tissue-specific *cis*-expression quantitative trait loci (*cis*-eQTL) were queried from the GTEx Portal (http://www.gtexportal.org/home/) to evaluate the effects of the SNPs on gene expressions in different human tissues. Data were obtained from the Genotype-Tissue Expression (GTEx) Portal on 4 May 2020. 

### 2.6. Phenome-Wide Association Study (PheWAS)

A phenome-wide association study (PheWAS) was conducted to screen the association between *WDR4* genetic variation and common traits. Imputed Axiom Genome-Wide TWB 2.0 Array data from 68,948 individuals were obtained from the Taiwan Biobank. The data were subjected to quality control based on a variant call rate (>98%), sex check, sample call rate (>98%), Hardy–Weinberg equilibrium (HWE; *p* > 10^−6^), heterozygosity (within the mean ± 3 standard deviations (SDs)), and identical by descent (IBD) check (<0.1875). After quality control and extracting common variants (with minor allele frequencies of >0.05), 3,982,815 variants and 59,448 individuals remained. Furthermore, 105 variants of the *WDR4* gene were extracted to conduct the PheWAS with 87 phenotypes using Plink v1.9 (www.cog-genomics.org/plink/1.9/) [15,16].

### 2.7. Cell Culture 

The mouse embryonic fibroblasts (MEF) used in this study were purchased from mWh heterozygous mice (+/−) [17] and maintained in DMEM (GIBCO, 11995) contained with 10% FBS, 1% Pen/Strep (GIBCO, 15140122). The 3 × 10^5^ cells were seeded in 6 cm culture dish for 24 h before treated with H_2_O_2_.

### 2.8. Western Blotting

To detect the reactive oxygen species (ROS)-caused DNA damage, cells were treated with 100 μM H_2_O_2_ for 24 h. Cells were then lysed, and the total proteins were analyzed by 10% SDS–polyacrylamide gel. After running the gel, proteins were transferred to a PVDF membrane and blocked with 5% non-fat dry milk for 1 h at RT. Membranes were then washed with PBST (PBS contained with 0.1% Tween 20) and incubated with primary antibodies in 1:1000 dilutions overnight at 4 °C; second antibodies were incubated in 1:5000 dilutions for 1 h at RT. The peptides LKKKRQRSPFPGSPEQTK were synthesized to detect mouse Wdr4 protein (mWdr4). The antibody was purified by affinity chromatography with peptide antigens before use [17]. The protein band intensities were detected by an ECL-plus detection system (GE Healthcare, RPN2235).

### 2.9. Statistical Analysis

For the PheWAS, plink v1.9 was used for analysis (www.cog-genomics.org/plink/1.9/) [15,16]. Furthermore, the simpleM method was used for variants (18 independent variants after correction), and Pearson’s correlation was used for phenotypes (76 independent phenotypes after correction with r^2^ > 0.7) to adjust the number of independent tests for multiple testing correction in PheWAS [18]. In this study, tests with *p* < 3.65 × 10^−5^ (0.05/18 × 76) were regarded as significant for phenome-wide associations. For the *WDR4* genetic association study with male infertility, R 3.2.0 (http://www.r-project.org) was used for the statistical analysis. HWE of SNPs was evaluated using a Chi-squared test. Magnitudes of associations between SNPs and semen quality parameters were examined through a linear regression analysis under a recessive model. Age, total testosterone, follicle-stimulating hormone (FSH), and luteinizing hormone (LH) were included as covariates in the regression model. *p* < 0.05 was considered statistically significant.

## 3. Results

### 3.1. PheWAS for WDR4 Variants in A Taiwanese Population

To further explore the roles of *WDR4* in human traits, a PheWAS was conducted to investigate associations between *WDR4* variants and common traits in a Taiwanese population. The associations between 105 variants of *WDR4* gene and 87 phenotypes were tested through the PheWAS. A variety of associations between the *WDR4* genetic variants and the common phenotypes were identified at the significance level of *p* < 0.05. However, none of them remained significant after multiple testing correction (Figure 2). 

### 3.2. Participant Characteristics

Eighty-one males were recruited from the infertility clinic. Participants were aged 28~51 years, with a mean age of 36 years. Men with oligozoospermia, asthenozoospermia, and teratozoospermia accounted for 15%, 27%, and 7%, respectively. Their clinical characteristics are summarized in Table 1. Furthermore, correlation coefficient of age, serum hormones, and semen parameters were shown by Spearman’s rank correlation. In agreement with the previous study, our data demonstrated morphology decline with age, whereas follicle-stimulating hormone (FSH) levels rose (Appendix A) [19].

### 3.3. Associations between WDR4 Variants and Oligozoospermia

The effects of four SNPs on total sperm number were first evaluated. There were no statistically significant differences in genotypic distributions between oligozoospermia cases and the non-oligozoospermia group (Table 2).

### 3.4. Associations between WDR4 Variants and Asthenozoospermia

Next, the effects of four SNPs on total sperm motility were analyzed. As shown in Table 3, the genotypic distribution of rs465663 differed between asthenozoospermia and non-asthenozoospermia groups. The proportions of TT and TC genotypes were higher in asthenozoospermia compared to non-asthenozoospermia (*p* = 0.025).

### 3.5. Associations between WDR4 Variants and Teratozoospermia

The effects of four SNPs on normal sperm morphology were also examined. As shown in Table 4, no statistically significant differences in genotypic distributions were observed between teratozoospermia cases and the non-teratozoospermia group.

### 3.6. Functional Annotation from GTEx Portal for rs465663

The profile of *WDR4* gene expression in various tissues was determined by the GTEx portal database (Figure 3). In the genetic association analysis, rs465663, an intronic variant of *WDR4*, was associated with asthenozoospermia. To further elucidate the possible functions of rs465663, *cis*-eQTL results were retrieved from the GTEx portal. rs465663 could affect the expression level of several genes in different tissue types. Subjects carrying the T allele showed a lower expression level of *WDR4* in testes, whole blood, and esophageal mucosal tissue (Appendix A). 

### 3.7. Effects of WDR4 in DNA Fragmentation through γH2AX Expression

Previous studies showed that the sperm DNA fragmentation level was increased by ROS stress and was correlated with sperm motility [5,6]. The higher expression level of γH2AX (biomarker of DNA fragmentation level) was revealed to correlate with male infertility [20]. To investigate the role of *WDR4* in male infertility, the MEF (mouse embryonic fibroblasts) cells were isolated from mwdr4 heterozygous (+/−) mice, and the H_2_O_2_ was used to induce DNA fragmentation for mimicking oxidative stress. The mwdr4 and the γH2AX protein expression levels were confirmed with western blotting (Figure 4). The results showed that the mWdr4 protein expression level in the mWdr4 heterozygous MEF group was low. Importantly, the H_2_O_2_-induced γH2AX protein level was highly increased in the mwdr4 heterozygous MEF group compared to the wild type MEF group. These results highlighted an important role of *WDR4* in male infertility.

## 4. Discussion

Wuho is a member of the evolutionarily conserved WD repeat protein family that is expressed by the genes *wuho* in Drosophila, *TRM82* in yeast, and *WDR4* in humans [17]. The *WDR4* domains usually contain four to eight repeating sequences, which are separated by approximately 40 amino acids. Each repeat consists of two sites, a poorly conserved site and a well-conserved site [21]. Wu at al. reported that the lack of *WDR4* function is associated with dramatic germline-specific phenotypes by arresting the spermatogenesis at the spermatid elongating stage [13]. The study indicated that approximately 20% of the ovarioles in *WDR4* mutant female have apparent defects in oogenesis with an over-proliferation of cystocytes. Additionally, *WDR4* is associated with germline cell development through cytosolic tRNA modifications [22].

Male infertility is characterized (manifested) by low sperm production (oligozoospermia), reduced sperm motility (asthenozoospermia), and abnormal sperm morphology (teratozoospermia). In this study, we wanted to seek out whether any genetic variants associate with the clinical outcomes. Importantly, the analysis using continuous traits showed consistent results with dichotomous models (Appendix A). Asthenozoospermia and oligozoospermia are the most common factors responsible for male infertility [23]. In line with this, our study indicated that asthenozoospermia comprised the most cases followed consecutively by oligozoospermia and teratozoospermia. Here, we identified variants of the *WDR4* gene that related to male infertility. rs465663 significantly associated with the susceptibility to asthenozoospermia. The variation in rs465663 was located in the intronic region of chromosome 21, while proportions of the TT and the TC genotypes of rs465663 were higher than the CC genotype in cases with asthenozoospermia. Interestingly, the C allele is the minor allele of rs465663. The distribution of the minor allele of rs465663 in current study was lower than those reported in other populations, including Asians, Europeans, and Americans (Appendix A). Trends of infertility globally showed that two populations of Europeans, represented by Central and Eastern Europe (8~12%), and Americans, represented by North America (4.5~6%), were higher compared to Sub-Saharan Africans (2.5~4.8%); unfortunately, those studies did not provide the percentage of Asians, possibly due to underreporting [24]. 

Our findings also emphasized that functional annotations through a bioinformatic approach using the GTEx portal revealed that subjects had lower expression of the TT genotype of *WDR4* than the CC and the TC genotypes in several human tissues, including testes and whole blood. Meanwhile, the major allelic frequency of rs465663 of *WDR4* had lower expression in testes. This result implied that rs465663 might influence expression of the *WDR4* gene, especially in testes. Disease-related male infertility can be caused by testicular deficiencies and spermatogenesis [4]. This evidence provided additional clues that variations of the *WDR4* gene may affect the testes, including their production of sperm (Figure 3). Furthermore, we noticed that ROS-induced DNA fragmentation level was significantly increased in the low *WDR4* group (Figure 4). The results further support the possibility that lower expression of *WDR4* in males resulted in infertility.

Several phenotypes are associated with *WDR4* variants, such as rs370189685, which was correlated with the fasting plasma glucose level [25]. A study from *Drosophila* germ cells suggested that *WDR4* is a regulator of Mei-p26, and it interacts with *TRIM32* to control tissue homeostasis in other stem cell systems [26]. Other genes related to male infertility were identified to associate with azoospermia (e.g., *SOHLH1* [27], *SYCP3* [28], and *TEX11* [29]). *WDR4* gene with a missense mutation was reported to be related to primordial dwarfism through m^7^G_46_ methylation, which impaired transfer (t)RNAs [30]. Sperm carries thousands of different RNAs [31]. Interestingly, according to the REACTOME database (R-HSA-6782315) [32], variations in rs465663 of the *WDR4* gene might involve in RNA metabolism and tRNA modifications in nuclei and cytosol. tRNAs play pivotal roles in protein synthetases. Mutations in tRNAs which modify enzymes are associated with human diseases, including cancer, type 2 diabetes (T2D), neurological disorders, and mitochondrion-linked disorders [33]. Furthermore, as reported by a previous study, tRNA modifications can affect proteostasis in humans [34]. A similar study supported that tRNA modifications in mice resulted in increased apoptosis in male germ cells and male infertility [35]. In addition, epigenomic assays are also essential for relating noncoding genetic variations to regulatory mechanisms underlying phenotypic changes, including genomic variations of the *WDR4* gene. The Encyclopedia of DNA Elements (ENCODE) database revealed that rs465663 is located in an intronic region with known histone modifications [32]. Subsequent evidence revealed that an epigenomic histone modification was involved in male fertility [36]. Some genes, such as *FAM50B* and *GNAS,* were reported to involve in the quality of sperm in asthenozoospermia through histone modification-type methylation [37]. Taken together, previous studies provided comprehensive integration between the identified genetic variants and male infertility. 

Our research is focusing on the genetic variants of *WDR4* in male fertility s. However, some limitations still exist in this study. First, although the subject number (68,948 individuals) for PheWAS was good, no significant physiological traits were found. Regarding the second cohort (male participants) from the hospital, the small sample size limits the statistical power. Thus, larger sample sizes with different populations are necessary to confirm our findings. Second, as the candidate gene approach was performed, the influence of variants in other unexamined genes cannot be ruled out. Since the pathophysiology of infertility in humans is complexly regulated by many signaling pathways, using whole genome sequencing technology may yield further insights into the genomic variations in infertility. 

## 5. Conclusions

This study offers important information related to genetic variants and expression level of the *WDR4* which might affect infertility in the cases of asthenozoospermia. However, further functional studies and larger sample sizes are required to validate the variants.

## Figures and Tables

**Figure 1 jpm-11-00760-f001:**
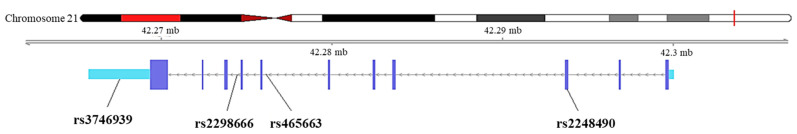
Graphic view of the genotyped *WDR4* gene.

**Figure 2 jpm-11-00760-f002:**
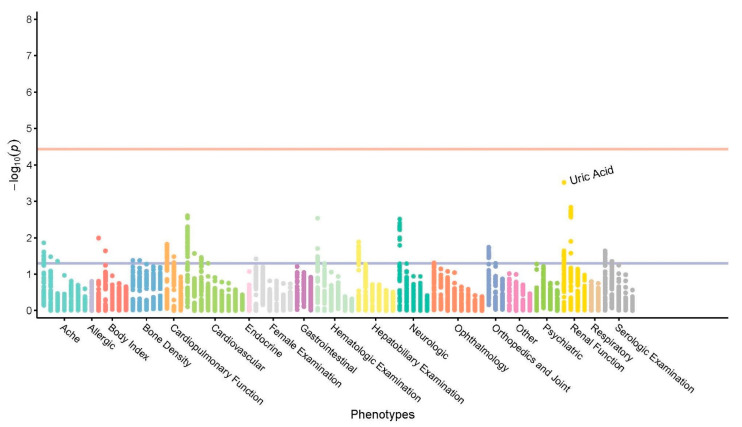
Phenome-wide scan for associations with *WDR4* variants. The blue line represents *p* = 0.05, and the red line represents the significance level for phenome-wide associations after multiple testing corrections (*p* = 3.65 × 10^−5^).

**Figure 3 jpm-11-00760-f003:**
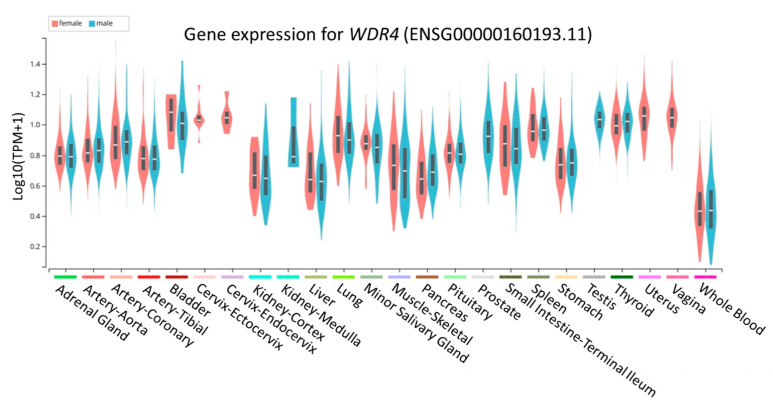
Profile of *WDR4* gene expression in various tissues. Blue represents the male gender and red the female gender.

**Figure 4 jpm-11-00760-f004:**
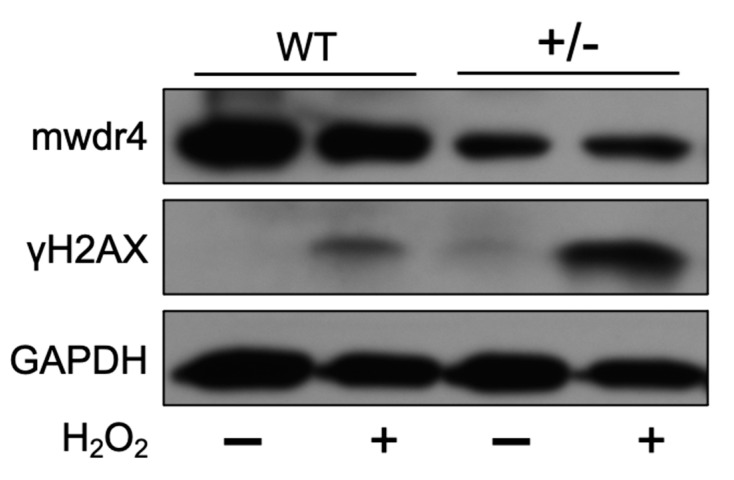
H_2_O_2_-induced γH2AX expression increased in wdr4 heterozygous (+/−) MEF cells. The MEFs cells were subcultured into 6 cm culture dish 24 h before being treated with 100 μM H_2_O_2_. The mwdr4, the γH2AX, and the GAPDH protein expression levels were determined by western blot after H_2_O_2_ treatment for 24 h.

**Table 1 jpm-11-00760-t001:** Characteristics of the 81 male participants.

Characteristic	Participants	Normal Range
Age (years) ^a^	36.21 ± 4.58	
Range (years)	27~51	
Semen analysis, no. (%)		
Oligozoospermia (total sperm number < 39 × 10^6^)		
Oligozoospermia	12 (15)	
Isolated	9	
Combined	3	
Non-oligozoospermia	69 (85)	
Asthenozoospermia (PR < 32%)		
Asthenozoospermia	22 (27)	
Isolated	10	
Combined	12	
Non-asthenozoospermia	59 (73)
Teratozoospermia (sperm with normal morphology <4%)		
Teratozoospermia	6 (7)	
Isolated	0	
Combined	6	
Non-teratozoospermia	75 (93)	
Serum hormone analysis ^a^		
Total testosterone (ng/mL)	4.87 ± 1.59	2.51~10.60
FSH (mIU/mL)	5.42 ± 4.24	1~14
LH (mIU/mL)	2.40 ± 1.59	1.5~9.2
SHBG (nmol/L)	28.88 ± 14.82	14.5~48.4
Inhibin B (pg/mL)	252.40 ± 162.44	—
TSH (μIU/mL)	1.63 ± 0.82	0.34~5.60
T3 (ng/dL)	112.20 ± 18.06	87~178
T4 (μg/dL)	8.15 ± 1.28	6.09~12.23
Free T4 (ng/dL)	0.90 ± 0.10	0.61~1.12
AMH (ng/mL)	12.51 ± 7.84	—
Zinc (μg/L)	1076.10 ± 217.26	800~1200

^a^ Mean ± standard deviation. PR, progressive motility; FSH, follicle-stimulating hormone; LH, luteinizing hormone; SHBG, sex hormone-binding globulin; TSH, thyroid-stimulating hormone; T3, triiodothyronine; T4, thyroxine; AMH, anti-müllerian hormone. — No optimal recommendation available for male.

**Table 2 jpm-11-00760-t002:** Associations between genetic variants of the *WDR4* gene and oligozoospermia.

SNP	Genotype	Cases		Non-Oligozoospermia		Recessive Model *p*-Value
		No.	%	No.	%	
rs2298666	GG	10	83.3	48	70.6	0.665
	GA	2	16.7	18	26.5	
	AA	0	0	2	2.9	
rs465663	TT	9	75.0	33	49.3	0.157
	TC	3	25.0	24	35.8	
	CC	0	0	10	14.9	
rs2248490	CC	8	66.7	30	43.5	0.095
	CG	4	33.3	29	42.0	
	GG	0	0	10	14.5	
rs3746939	AA	8	66.7	37	53.6	0.165
	AC	4	33.3	27	39.1	
	CC	0	0	5	7.2	

The *p*-value was adjusted for age, total testosterone, follicle-stimulating hormone, and luteinizing hormone. SNP, single-nucleotide polymorphism.

**Table 3 jpm-11-00760-t003:** Associations between genetic variants of the *WDR4* gene and asthenozoospermia.

SNP	Genotype	Cases		Non-Asthenozoospermia		Recessive Model *p*-Value
		No.	%	No.	%	
rs2298666	GG	17	81	41	69.5	0.399
	GA	4	19	16	27.1	
	AA	0	0	2	3.4	
rs465663	TT	15	71.4	27	46.6	0.025 *
	TC	6	28.6	21	36.2	
	CC	0	0	10	17.2	
rs2248490	CC	15	68.2	23	39	0.677
	CG	5	22.7	28	47.5	
	GG	2	9.1	8	13.6	
rs3746939	AA	15	68.2	30	50.8	0.513
	AC	6	27.3	25	42.4	
	CC	1	4.5	4	6.8	

The *p*-value was adjusted for age, total testosterone, follicle-stimulating hormone, and luteinizing hormone. * *p* < 0.05. SNP, single-nucleotide polymorphism.

**Table 4 jpm-11-00760-t004:** Associations between genetic variants of the *WDR4* gene and teratozoospermia.

SNP	Genotype	Cases		Non-Teratozoospermia		Recessive Model *p*-Value
		No.	%	No.	%	
rs2298666	GG	5	83.3	53	71.6	0.759
	GA	1	16.7	19	25.7	
	AA	0	0	2	2.7	
rs465663	TT	5	83.3	37	50.7	0.395
	TC	1	16.7	26	35.6	
	CC	0	0	10	13.7	
rs2248490	CC	5	83.3	33	44	0.154
	CG	1	16.7	32	42.7	
	GG	0	0	10	13.3	
rs3746939	AA	4	66.7	41	54.7	0.186
	AC	2	33.3	29	38.7	
	CC	0	0	5	6.7	

The *p*-value was adjusted for age, total testosterone, follicle-stimulating hormone, and luteinizing hormone. SNP, single-nucleotide polymorphism.

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
