# Peer review of "Genetic Association of the Functional WDR4 Gene in Male Fertility"

_jpm, 2021, doi:10.3390/jpm11080760_

Round 1
Reviewer 1 Report
In the introduction, the authors reference CF mutations and their association with impaired spermatogenesis and impaired sperm quality. This is a relatively poor choice of an example, as the relationship of CF mutations and male infertility is weak (beyond structural anomalies in the Wolffian tract.)
Author Response
Respond to Reviewer 1 comment:
Q1: In the introduction, the authors reference CF mutations and their association with impaired spermatogenesis and impaired sperm quality. This is a relatively poor choice of an example, as the relationship of CF mutations and male infertility is weak (beyond structural anomalies in the Wolffian tract.)
Ans: We sincerely thank the reviewer's comment. As suggested by the reviewer, we have removed the references of CF mutations and have modified the introduction part as stated by the following sentences (page 2, lines 59-65): “In addition, HSFY, which is related to the heat-shock transcription factor (HSF) family that found on chromosome Yq, has a long open reading frame carrying an HSF-type DNA binding domain involved in azoospermia. Some point mutations in the testis-specific HSFY gene family, such as microdeletions, may be a cause of unexplained cases of idiopathic male infertility [1]. Furthermore, a study evaluated the role of genetic variants in male infertility; There were three SNPs (rs4920566, rs11763979, and rs3741843) were associated with taste receptor genes (TASR) and male infertility [2]. Besides the genomic variants, the epigenetic modifications also play important roles in spermatogenesis and male infertility.”
Reference:
- Ozdemir O, Gul E, Kilicarslan H, Gokce G, Beyaztas FY, Ayan S, Sezgin I: SRY and AZF gene variation in male infertility: a cytogenetic and molecular approach. Int Urol Nephrol 2007, 39(4):1183-1189.
- Gentiluomo M, Crifasi L, Luddi A, Locci D, Barale R, Piomboni P, Campa D: Taste receptor polymorphisms and male infertility. Human Reproduction 2017, 32(11):2324-2331.

Reviewer 2 Report
Reviewed manuscript is interesting and highlights the association of the WDR-4 and male fertility. Generally manuscript is well written, but there are few points that deserve further clarification:
- Section 2. Semen analysis: Semen analysis should be performed according to the WHO 2010 guideline. Please provide this information. Most important, WHO define asthenozoospermia as <32% progressives sperm motility, this parameter is more important than total sperm motility (WHO 2010: page 226 Table A1.3: ‘percentage of progressively motile (PR) spermatozoa below the lower reference limit’), similarly in case of oligozoospermia WHO give different definition, more important is total sperm count than concentration (WHO 2010: total number (or concentration, depending on outcome reported)* of spermatozoa below the lower reference limit; *Preference should always be given to total number, as this parameter takes precedence over concentration). Because the authors divide the group according to asthenozoospermia, oligozoospermia or teratozoospermia it is extremely important to assign patients the appropriate category of semen disorders.
- Whether the men had an isolated asthenozoospermia, oligozoospermia or teratozoospermia? If yes, it should be clearly written, if not authors should use term: non-asthenozoospermia group (men with other disorders than asthenozoospermia: oligoteratozoospermia, teratozoospermia, oligozoospermia…); non-oligozoospermia group and non-teratozoospermia group. Unfortunetly, it the subjects didn’t have isolated seminological disorders, these changes should be implemented consistently throughout the rest of the manuscript (tables and statistics).
- Section 3.7. Effects of WDR4 in DNA fragmentation: The authors did not investigate directly DNA fragmentation but the protein expression. In my opinion, this section requires
- Section 4. Discussion: sentences from line 256 to 264 about epidemiology of infertility should be compiled with sentences in introduction. This background does not fit in with the discussion. The authors should focus on discussing their own results, comparing them with published data and explaining the role of the studied gene in male fertility.
- As mentioned above, in discussion authors should describe in detail the role of the WDR-4 in spermatogenesis. I suggest create a separate subsection in which the role of WDR-4 in spermatogenesis and the pathomechanism associated with mutations of this gene will be described.
- Conclusions: It is worth mentioning that in the case of infertile men, when intracytoplasmic sperm injection is used as a method of infertility treatment, there is a risk of passing these changes on to the next generation, which patients should be informed about.
Author Response
Respond to Reviewer 2 comment:
Q1: Section 2. Semen analysis: Semen analysis should be performed according to the WHO 2010 guideline. Please provide this information. Most important, WHO define asthenozoospermia as <32% progressives sperm motility, this parameter is more important than total sperm motility (WHO 2010: page 226 Table A1.3: ‘percentage of progressively motile (PR) spermatozoa below the lower reference limit’), similarly in case of oligozoospermia WHO give different definition, more important is total sperm count than concentration (WHO 2010: total number (or concentration, depending on outcome reported) of spermatozoa below the lower reference limit; Preference should always be given to total number, as this parameter takes precedence over concentration). Because the authors divide the group according to asthenozoospermia, oligozoospermia or teratozoospermia it is extremely important to assign patients the appropriate category of semen disorders.
Whether the men had an isolated asthenozoospermia, oligozoospermia or teratozoospermia? If yes, it should be clearly written, if not authors should use term: non-asthenozoospermia group (men with other disorders than asthenozoospermia: oligoteratozoospermia, teratozoospermia, oligozoospermia..); non-oligozoospermia group and non-teratozoospermia group. Unfortunately, if the subjects didn’t have isolated seminological disorders, these changes should be implemented consistently throughout the rest of the manuscript (tables and statistics).
Ans: Thanks for your valuable comments and suggestions. According to your suggestions, we have modified the definition of asthenozoospermia (PR<32%) and oligozoospermia (total sperm number <39x106). The data are shown in table 2 and 3. The paragraphs have been amended in the methods and results sections. For clarification, we showed sample numbers for both isolated and combined defects among groups in Table 1. In addition, we changed the term “controls” to “non-oligozoospermia”, “non-asthenozoospermia”; and “non-teratozoospermia” throughout the revised manuscript. We revised accordingly in the method section part 2.2 (page 2-3, lines 92-108). The following paragraph is the revised version. “Semen samples were examined after liquefaction. Semen analysis was conducted separately by two experienced technicians in the reproductive laboratory. Semen parameters, including sperm count, sperm motility, and sperm morphology were evaluated. The World Health Organization (WHO) 2010 recommendations for semen analysis were followed. A semen sample (10 µL) was first transferred to a MAKLER counting chamber (Irvine Scientific, Santa Ana, CA, USA) to measure the sperm count. Oligozoospermia was defined as a number of sperms <39x106 spermatozoa. The motility of each spermatozoon was graded into three categories: progressive motility (PR), non-progressive motility (NP), and immotile spermatozoa. Asthenozoospermia was defined as PR<32%. For sperm morphological evaluation, each spermatozoon was classified into eight groups as follows: normal sperm (NS), amorphous head (AH), duplicated head (DH), large head (LH), small head (SH), normal head with other defect (NH), tapering head (TH), and other defect (OD). Teratozoospermia was defined as <4% of morphologically normal spermatozoa. In addition, combined oligozoospermia defined as participants with sperm number less than 39x106 spermatozoa and other defects (astheno- and/or teratozoospermia), which rationale applies to other two sperm characteristics (combined asthenozoospermia and combined teratozoospermia)”.
Furthermore, we also adjusted the sentence in the result section page 6 (lines 207-209). “The effects of four SNPs on total sperm number were first evaluated. There were no statistically significant differences in genotypic distributions between oligozoospermia and non-oligozoospermia groups (Table 2)”. And page 7, lines 214-217. “Next, the effects of four SNPs on total sperm motility were analyzed. As shown in Table 3, the genotypic distribution of rs465663 differed between asthenozoospermia and non-asthenozoospermia groups. The proportions of TT and TC genotypes were higher in asthenozoospermia compared to non-asthenozoospermia (p=0.006)”.
Q2: Section 3.7. Effects of WDR4 in DNA fragmentation: The authors did not investigate directly DNA fragmentation but the protein expression. In my opinion, this section requires.
Ans: We sincerely thank the reviewer's comments. Yes. We did not investigate directly DNA fragmentation but the protein expression. Therefore, we improve the sentences as follows: “Previous studies showed that the sperm DNA fragmentation level was increased by ROS stress, and was correlated with sperm motility [5, 6]. The higher expression level of γH2AX (biomarker of DNA fragmentation level) has been revealed to correlate with male infertility [20]. To investigate the role of WDR4 in male infertility, the MEF (mouse embryonic fibroblasts) cells were isolated from mwdr4 heterozygous (+/-) mice, and the H2O2 was used to induce DNA fragmentation for mimicking oxidative stress. The mwdr4 and γH2AX protein expression level were confirmed with western blotting (Figure 4). The results showed that the mWdr4 protein expression level in mWdr4 heterozygous MEF group is low. Importantly, the H2O2-induced γH2AX protein level was highly increased in mwdr4 heterozygous MEF group compare to wild type MEF group. These results highlighted an important role of WDR4 in male infertility.”
Q3: Section 4. Discussion: sentences from lines 256 to 264 about epidemiology of infertility should be compiled with sentences in introduction. This background does not fit in with the discussion. The authors should focus on discussing their own results, comparing them with published data and explaining the role of the studied gene in male fertility.
Ans: Thank you very much for reviewer’s suggestions. We already adjusted and compile the epidemiology of infertility in the Introduction part (Page 1-2, lines 46-49). “Infertility is a global health issue affecting 8%~12% of couples worldwide, and male infertility factors are responsible for about 20%~30% of infertility cases. Approximately 7% of the male population is affected by infertility”.
Q4: As mentioned above, in discussion authors should describe in detail the role of the WDR-4 in spermatogenesis. I suggest creating a separate subsection in which the role of WDR-4 in spermatogenesis and the pathomechanism associated with mutations of this gene will be described.
Ans: We sincerely thank the reviewer for the time taken to review our work and the important suggestion given. References of WDR4 are very limited. The function of WDR4 has been mentioned in the separate subsection of discussion part (Page 9, line 260-269). “Wuho is a member of the evolutionarily conserved WD repeat protein family that is expressed by the genes wuho in Drosophila, TRM82 in yeast, and WDR4 in humans [1]. WD40 domains usually contain four to eight repeating sequences, which are separated by approximately 40 amino acids. Each repeat consists of two sites, poorly conserved site and well-conserved site [2]. Jianhong Wu at al. reported that the lack of WDR4 function is associated with dramatic germline-specific phenotypes by arresting the spermatogenesis at spermatid elongating stage [3]. The study also mentioned that approximately 20% of the ovarioles in WDR4 mutant female have apparent defects in oogenesis with an over-proliferation of cystocytes. Additionally, WDR4 also has associated with germline cell development through cytosolic tRNA modifications [4]”.
Reference:
- Cheng IC, Chen BC, Shuai H-H, Chien F-C, Chen P, Hsieh T-s: Wuho Is a New Member in Maintaining Genome Stability through its Interaction with Flap Endonuclease 1. PLoS biology 2016, 14(1):e1002349.
- Riedl SJ, Salvesen GS: The apoptosome: signalling platform of cell death. Nature Reviews Molecular Cell Biology 2007, 8(5):405-413.
- Wu J, Hou JH, Hsieh TS: A new Drosophila gene wh (wuho) with WD40 repeats is essential for spermatogenesis and has maximal expression in hub cells. Developmental biology 2006, 296(1):219-230.
- Tahmasebi S, Khoutorsky A, Mathews MB, Sonenberg N: Translation deregulation in human disease. Nature Reviews Molecular Cell Biology 2018, 19(12):791-807.
Q5: Conclusions: It is worth mentioning that in the case of infertile men, when intracytoplasmic sperm injection is used as a method of infertility treatment, there is a risk of passing these changes on to the next generation, which patients should be informed about.
Ans: Thank you very much for the reviewer's suggestions. This study indicated that genetic variants and expression level of the WDR4 gene is associated with infertility in males in cases of asthenozoospermia. It is ideal to translate medical research into clinical usage. However, the sample size is still small in this study. Larger sample size in different populations are required to confirm the polymorphism. Then, the clinical advices should be provided in the future.

Reviewer 3 Report
The manuscript entitled "Genetic association of the functional WDR-4 gene in male fertility" is well written and structured. The authors have explored the possible association between WDR-4 gene variants and several male infertility outcomes.
Although I appreciate the quality of this work, in my opinion, some clarifications and changes are still needed.
- Page 3, lines 115-116. The authors must explain the exact name of the tool used to identify tagging SNPs and the selection parameters.
- Page3, line 130. Add the date of access to GTEx and the version number consulted. Just the access date is present in Sup.table2 footnote; this information is needed in the main text.
- Page 3, line 141. To increase the author's analyses' reproducibility, I recommended adding a reference or the specific PLINK commands used for PheWAS.
- Page 4, lines 172-174. I can suppose that the authors adopted just the recessive model of inheritance and excluded the codominant e dominant model in the statistical analyses due to the very few cases.
In this study, the selected tagging SNPs have a MAF >10%, and the largest number of cases was 30 for Asthenozoospermia. I would suggest that the authors performing a power calculation and mentioned the power in the main text.
As also admitted by the authors, the number of subjects in the study is the major limitation. I agree with this statement; these numbers make any conclusion speculative regarding studying the association between genetic variants and any outcome. The authors could re-perform the analyses changing the setting of outcomes from a dichotomous variable to a continuous variable and from a logistic to a generalised linear model (GLM) analyses of the association, like suggested in this manuscript PMID: 29040583 DOI: 10.1093/humrep/dex305.
- Page 4, lines 172-174. Were tested the association of age, total testosterone, FSH, LH with the outcomes? These variables are included in the analyses due to their known association with male infertility but could be interestingly shown their association in the study population used for this manuscript.
- Page 9, lines 278-290. Reword this sentence. The phrase formulated in this way misunderstands the meaning of eQTL. "in the presence of a specific genotype, an altered expression of a gene X is observed". We can speak of the presence or non-presence of a genotype, but a genotype is not expressed, i.e. a gene is expressed.
Finally, the authors should emphasise that their results are less than suggestive, given the small number of samples that makes this study the victim of numerous errors due to statistical fluctuations.
So the results presented in this current form may be a mild suggestion for future studies done on larger populations.
Furthermore, the authors throughout the manuscript, from introduction to conclusion, cite and compare the WDR-4 gene of Homo sapiens, Drosophila and Mouse. Personally, it is little functional to support this study the comparison with orthologous genes beyond the introduction. However, if the authors maintain this structure of the manuscript, I believe a section is necessary where the WDR-4 gene is compared in the three species, indicating the degree of conservation and identity, a comparison that can be done with the BLAST tool https://blast.ncbi.nlm.nih.gov/Blast.cgi.
Author Response
Respond to Reviewer 3 comment:
Q1: Page 3, lines 115-116. The authors must explain the exact name of the tool used to identify tagging SNPs and the selection parameters.
Ans: Thanks for the reviewer’s comments. Following your suggestions, the paragraph have been added in the method section (2.4. Genotyping of SNPs in the WDR4 gene) as below: “Tagging (t)SNPs with a minimum allele frequency (MAF) of >10% in a Beijing Han Chinese (CHB) population were selected through UCSC (http://genome.ucsc.edu) and HapMap vers. 2010-08_phase II + III (http://hapmap.ncbi.nlm.hig.gov/). Haploview 4.2 was applied for tSNP selection”.
Q2: Page3, line 130. Add the date of access to GTEx and the version number consulted. Just the access date is present in Sup.table2 footnote; this information is needed in the main text.
Ans: Thanks for your kind reminder. We added the accession date of the GTEx as mentioned in page 3, line 136-137: “Data were obtained from the GTEx (V8) Portal on May 4, 2020”.
Q3: Page 3, line 141. To increase the author's analysis' reproducibility, I recommended adding a reference or the specific PLINK commands used for PheWAS.
Ans: We sincerely thank the reviewer's comment. Regarding to PLINK commands or reference, we followed the guidance from PLINK website (https://www.cog-genomics.org/plink/). The reference of PLINK was added in the revised manuscript.
Q4: Page 4, lines 172-174. I can suppose that the authors adopted just the recessive model of inheritance and excluded the codominant and dominant model in the statistical analyses due to the very few cases. In this study, the selected tagging SNPs have a MAF >10%, and the largest number of cases was 30 for Asthenozoospermia. I would suggest that the authors perform a power calculation and mention the power in the main text. As also admitted by the authors, the number of subjects in the study is the major limitation. I agree with this statement; these numbers make any conclusion speculative regarding studying the association between genetic variants and any outcome. The authors could re-perform the analyses changing the setting of outcomes from a dichotomous variable to a continuous variable and from a logistic to a generalised linear model (GLM) analyses of the association, like suggested in this manuscript PMID: 29040583 DOI: 10.1093/humrep/dex305.
Ans: Thanks for your valuable comments and suggestions. Indeed, number of the subjects is the major limitation of our study, which may lead to false negative findings. In this study, the nonsignificant findings in the association between WDR4 variations and oligozoospermia as well as teratozoospermia may be a victim of this. Furthermore, we have performed the analysis on continuous outcome variable in this revision. Results demonstrated no statistically significant association of WDR4 variants with sperm counts and morphology. The variants of WDR4 associated with the sperm motility, which is consistent with our findings with dichotomous outcomes. All results have been added in Supplementary Table S3-S5., and we also added sentences in the revised manuscript. “Here we performed the analysis by using dichotomous model. However, we also analyzed the data using continuous model. The results are consistent with our findings by utilizing dichotomous model (Table S3-5)”.
|
Table S3. Associations between genetic variants of the WDR4 gene and total sperm number |
||||||||||||||||||||||||||||||||||||||||||||||||||||||||||||||||||||||||||||||||||||||||||||||||||||||||||||||||||||||||||||||||||||||||||||||||||||||||||||||||||||||||||||||||||||||||||||||||||||||||||||||||||||||||||||||||||||||||||||||||||||||||||||||||||||||||||||||||||||||||||||||||
|
SNP |
Genotype |
No |
Value |
|
p-value |
|||||||||||||||||||||||||||||||||||||||||||||||||||||||||||||||||||||||||||||||||||||||||||||||||||||||||||||||||||||||||||||||||||||||||||||||||||||||||||||||||||||||||||||||||||||||||||||||||||||||||||||||||||||||||||||||||||||||||||||||||||||||||||||||||||||||||||||||||||||||||||
|
Mean |
S.E. |
Dominant |
Recessive |
Additive |
||||||||||||||||||||||||||||||||||||||||||||||||||||||||||||||||||||||||||||||||||||||||||||||||||||||||||||||||||||||||||||||||||||||||||||||||||||||||||||||||||||||||||||||||||||||||||||||||||||||||||||||||||||||||||||||||||||||||||||||||||||||||||||||||||||||||||||||||||||||||||||
|
rs2298666 |
GG |
58 |
221.50 |
25.85 |
|
0.924 |
0.801 |
0.995 |
||||||||||||||||||||||||||||||||||||||||||||||||||||||||||||||||||||||||||||||||||||||||||||||||||||||||||||||||||||||||||||||||||||||||||||||||||||||||||||||||||||||||||||||||||||||||||||||||||||||||||||||||||||||||||||||||||||||||||||||||||||||||||||||||||||||||||||||||||||||||
|
|
GA |
20 |
217.81 |
47.97 |
|
|
|
|
||||||||||||||||||||||||||||||||||||||||||||||||||||||||||||||||||||||||||||||||||||||||||||||||||||||||||||||||||||||||||||||||||||||||||||||||||||||||||||||||||||||||||||||||||||||||||||||||||||||||||||||||||||||||||||||||||||||||||||||||||||||||||||||||||||||||||||||||||||||||
|
|
AA |
2 |
254.00 |
76.00 |
|
|
|
|
||||||||||||||||||||||||||||||||||||||||||||||||||||||||||||||||||||||||||||||||||||||||||||||||||||||||||||||||||||||||||||||||||||||||||||||||||||||||||||||||||||||||||||||||||||||||||||||||||||||||||||||||||||||||||||||||||||||||||||||||||||||||||||||||||||||||||||||||||||||||
|
rs465663 |
TT |
42 |
193.83 |
29.88 |
|
0.314 |
0.427 |
0.279 |
||||||||||||||||||||||||||||||||||||||||||||||||||||||||||||||||||||||||||||||||||||||||||||||||||||||||||||||||||||||||||||||||||||||||||||||||||||||||||||||||||||||||||||||||||||||||||||||||||||||||||||||||||||||||||||||||||||||||||||||||||||||||||||||||||||||||||||||||||||||||
|
|
TC |
27 |
228.79 |
35.89 |
|
|
|
|
||||||||||||||||||||||||||||||||||||||||||||||||||||||||||||||||||||||||||||||||||||||||||||||||||||||||||||||||||||||||||||||||||||||||||||||||||||||||||||||||||||||||||||||||||||||||||||||||||||||||||||||||||||||||||||||||||||||||||||||||||||||||||||||||||||||||||||||||||||||||
|
|
CC |
10 |
290.92 |
74.94 |
|
|
|
|
||||||||||||||||||||||||||||||||||||||||||||||||||||||||||||||||||||||||||||||||||||||||||||||||||||||||||||||||||||||||||||||||||||||||||||||||||||||||||||||||||||||||||||||||||||||||||||||||||||||||||||||||||||||||||||||||||||||||||||||||||||||||||||||||||||||||||||||||||||||||
|
rs2248490 |
CC |
38 |
213.36 |
34.27 |
|
0.734 |
0.549 |
0.974 |
||||||||||||||||||||||||||||||||||||||||||||||||||||||||||||||||||||||||||||||||||||||||||||||||||||||||||||||||||||||||||||||||||||||||||||||||||||||||||||||||||||||||||||||||||||||||||||||||||||||||||||||||||||||||||||||||||||||||||||||||||||||||||||||||||||||||||||||||||||||||
|
|
CG |
33 |
242.80 |
35.85 |
|
|
|
|
||||||||||||||||||||||||||||||||||||||||||||||||||||||||||||||||||||||||||||||||||||||||||||||||||||||||||||||||||||||||||||||||||||||||||||||||||||||||||||||||||||||||||||||||||||||||||||||||||||||||||||||||||||||||||||||||||||||||||||||||||||||||||||||||||||||||||||||||||||||||
|
|
GG |
10 |
218.60 |
48.40 |
|
|
|
|
||||||||||||||||||||||||||||||||||||||||||||||||||||||||||||||||||||||||||||||||||||||||||||||||||||||||||||||||||||||||||||||||||||||||||||||||||||||||||||||||||||||||||||||||||||||||||||||||||||||||||||||||||||||||||||||||||||||||||||||||||||||||||||||||||||||||||||||||||||||||
|
rs3746939 |
AA |
45 |
209.71 |
28.78 |
|
0.503 |
0.531 |
0.428 |
||||||||||||||||||||||||||||||||||||||||||||||||||||||||||||||||||||||||||||||||||||||||||||||||||||||||||||||||||||||||||||||||||||||||||||||||||||||||||||||||||||||||||||||||||||||||||||||||||||||||||||||||||||||||||||||||||||||||||||||||||||||||||||||||||||||||||||||||||||||||
|
|
AC |
31 |
238.43 |
39.16 |
|
|
|
|
||||||||||||||||||||||||||||||||||||||||||||||||||||||||||||||||||||||||||||||||||||||||||||||||||||||||||||||||||||||||||||||||||||||||||||||||||||||||||||||||||||||||||||||||||||||||||||||||||||||||||||||||||||||||||||||||||||||||||||||||||||||||||||||||||||||||||||||||||||||||
|
|
CC |
5 |
295.60 |
80.12 |
|
|
|
|
||||||||||||||||||||||||||||||||||||||||||||||||||||||||||||||||||||||||||||||||||||||||||||||||||||||||||||||||||||||||||||||||||||||||||||||||||||||||||||||||||||||||||||||||||||||||||||||||||||||||||||||||||||||||||||||||||||||||||||||||||||||||||||||||||||||||||||||||||||||||
|
The p-value was adjusted for age, total testosterone, follicle-stimulating hormone, and luteinizing hormone. SNP, single-nucleotide polymorphism. S.E., standard error.
|
||||||||||||||||||||||||||||||||||||||||||||||||||||||||||||||||||||||||||||||||||||||||||||||||||||||||||||||||||||||||||||||||||||||||||||||||||||||||||||||||||||||||||||||||||||||||||||||||||||||||||||||||||||||||||||||||||||||||||||||||||||||||||||||||||||||||||||||||||||||||||||||||
Q5: Page 4, lines 172-174. Were tested the association of age, total testosterone, FSH, LH with the outcomes? These variables are included in the analyses due to their known association with male infertility but could be interestingly shown their association in the study population used for this manuscript.
Ans: Thanks for your valuable comments and suggestions. We did the Spearman's rank correlation coefficient of age, serum hormones and semen parameters that showed as below. We added this analysis in the revised manuscript. “Furthermore, the correlation coefficient of age, serum hormones, and semen parameters were showed by Spearman's rank correlation. In agreement with the previous study, our data demonstrated morphology decline with age, whereas follicle-stimulating hormone (FSH) levels rise (Figure S1) [1]”.
Figure S1. Spearman's rank correlation coefficient between age, serum hormones, and semen parameters
Reference:
- Pasqualotto FF, Sobreiro BP, Hallak J, Pasqualotto EB, Lucon AM: Sperm concentration and normal sperm morphology decrease and follicle-stimulating hormone level increases with age. BJU Int 2005, 96(7):1087-1091.
Q6: Page 9, lines 278-290. Reword this sentence. The phrase formulated in this way misunderstands the meaning of eQTL. "in the presence of a specific genotype, an altered expression of a gene X is observed". We can speak of the presence or non-presence of a genotype, but a genotype is not expressed, i.e. a gene is expressed. Finally, the authors should emphasise that their results are less than suggestive, given the small number of samples that makes this study the victim of numerous errors due to statistical fluctuations. So the results presented in this current form may be a mild suggestion for future studies done on larger populations.
Furthermore, the authors throughout the manuscript, from introduction to conclusion, cite and compare the WDR-4 gene of Homo sapiens, Drosophila and Mouse. Personally, it is little functional to support this study the comparison with orthologous genes beyond the introduction. However, if the authors maintain this structure of the manuscript, I believe a section is necessary where the WDR-4 gene is compared in the three species, indicating the degree of conservation and identity, a comparison that can be done with the BLAST tool https://blast.ncbi.nlm.nih.gov/Blast.cgi.
Ans: Thanks so much for reviewer’s suggestions. We already added your concern regarding to limitation this study in the part of discussion (Page 10, 328-333). “Our research contributes to the understanding of the WDR-4 gene's variations. However, some limitations still exist in this study; first, although the subject number (68,948 individuals) for PheWAS is good, none of significant physiological traits was found. Regarding the second cohort (male infertility) from hospital, small sample size limits the statistical power. Thus, a larger sample sizes with different populations are necessary to confirm these findings”.
Regarding to WDR4-degree conservation, we add some related information as below:
“Wuho is a member of the evolutionarily conserved WD repeat protein family that is expressed by the genes wuho in Drosophila, TRM82 in yeast, and WDR4 in humans [1]. The WDR4 domains usually contain four to eight repeating sequences, which are separated by approximately 40 amino acids. Each repeat consists of two sites, poorly conserved site and well-conserved site [2]”
Reference:
- Cheng IC, Chen BC, Shuai H-H, Chien F-C, Chen P, Hsieh T-s: Wuho Is a New Member in Maintaining Genome Stability through its Interaction with Flap Endonuclease 1. PLoS biology 2016, 14(1):e1002349.
- Riedl SJ, Salvesen GS: The apoptosome: signalling platform of cell death. Nature Reviews Molecular Cell Biology 2007, 8(5):405-413.
